# Research on Multi-Sensor Fusion Indoor Fire Perception Algorithm Based on Improved TCN

**DOI:** 10.3390/s22124550

**Published:** 2022-06-16

**Authors:** Yang Li, Yanmang Su, Xiangye Zeng, Jingyi Wang

**Affiliations:** 1School of Electronic Information Engineering, Hebei University of Technology, Tianjin 300401, China; yuuki19970617@163.com (Y.L.); zengxy@hebut.edu.cn (X.Z.); 2Tianjin Key Laboratory of Electronic Materials and Devices, School of Electronics and Information Engineering, Hebei University of Technology, Tianjin 300401, China; wangjingyi@hebut.edu.cn

**Keywords:** fire perception, multi-sensor fusion, trend extraction, TCN, AAP, SVM

## Abstract

Indoor fires cause huge casualties and economic losses worldwide. Thus, it is critical to quickly and accurately perceive the fire. In this work, an indoor fire perception algorithm based on multi-sensor fusion was proposed. Firstly, the sensor data features were fully extracted by improved temporal convolutional network (TCN). Then, the dimension of the extracted features was reduced by adaptive average pooling (AAP). Finally, the fire classification was realized by the support vector machine (SVM) classifier. Experimental results demonstrated that the proposed algorithm can improve accuracy of fire classification by more than 2.5% and detection speed by more than 15%, compared with TCN, back propagation (BP) neural network and long short-term memory (LSTM). In conclusion, the proposed algorithm can perceive the fire quickly and accurately, which is of great significance to improve the performance of the current fire prediction systems.

## 1. Introduction

In recent years, the development of electrical equipment and new organic materials has brought about huge fire hazards, resulting in huge casualties and economic losses worldwide. China reported 748,000 fires including 2225 injuries, 1987 deaths, and CNY 6.75 billion of direct property damage in 2021 [1]. In the above events, the most prominent point was that residential fires only occupied 34.5% of all fires, but up to 73.8% of all deaths. For indoor fires, the potential danger which causes larger casualties and economic losses will become more prominent as related technology and materials rapidly develop. In such circumstances, the study and solution of indoor fires need to be updated in order to reduce the various losses. Therefore, the research of indoor fire perception is quite necessary and urgent.

Many physical characteristics change in the early period of the fire, such as temperature, smoke, CO, and so on [2]. Thus, the aim of indoor fire perception is capturing corresponding changes quickly and accurately, and then responding. Sensor-based fire detectors have been widely used in various fields of fire detection, owing to convenient installation and real-time monitoring [3,4,5,6]. Fire detection technology has developed rapidly over the past 100 years since the first temperature-sensitive fire detector was designed and implemented in 1890 in the UK [7]. Nowadays, multi-sensor synergistic fire detectors are booming [8,9]. The sensors work together to more accurately capture the anomalous changes of various features during a fire, which greatly improves the performance of fire detectors. However, the multiple information sensed have complex non-linear relationships with each other, and the information is fused by using intelligent algorithms to make scientific decisions [10]. In view of the intelligent algorithms, numerous experts and scholars have conducted relative studies.

The prevailing multi-sensor data fusion methods for fire perception can be divided into two categories:
(1)The methods based on Bayesian estimation, statistics, and inference [11]: Wang et al. [12] proposed a fire detection system based on Kalman filter. The Kalman filter was used to fuse the data of sensors, then output the probabilities of no-fire, flaming, and smoldering into the system. Rachman et al. [13] used fuzzy logic rules to fuse data from various sensors in complex fire scenarios. Wang et al. [14] designed and performed modified hierarchical analysis to calculate the weight of each sensor. Then the method of multivariate weighted fusion was used to assess the probability of fire occurrence. Avgeris et al. [15,16] used edge computing technology to overcome the shortcomings of the fire perception system based on the Internet of Things, such as limited energy resources and lack of real-time processing computing ability. Therefore, the fire perception system with edge computing technology can quickly detect the fire situation and take appropriate measures. Real-time performance of the fire perception system was improved on account of the above computational simplicity methods.(2)The methods based on artificial intelligence: Zheng et al. [17,18] proposed the method that combined intelligent algorithms with a BP neural network to fuse multi-sensor data for fire perception. Similarly, Deng et al. [19] used a weight adaptive adjustment method to combine a BP neural network in order to achieve good performance. The described methods ameliorated the problem that BP neural networks tended to fall into local minima during the training process. Baek et al. [20] used a dynamic time warping (DTW) algorithm to evaluate the similarity of sensor data before and after a fire. In addition, the k-out-of-p rule based on p-channel sensor data was proposed to make decisions adaptively. Sun et al. [21] used LSTM to extract multi-sensor features and output the probabilities of no-fire, flaming, and smoldering. Moreover, a decision tree algorithm was used to produce the final fire detection result. The characteristic of the above method was to make scientific decisions while fusing sensor data, which enhanced the robustness of the fire perception system.

Each of the above solutions has its advantages for fire perception, but several deficiencies ought to be studied and solved. To begin with, the time dimension information of sensor data was not sufficiently considered. In the early stage of a fire, the sensor data showed a steady upward or downward trend in the long term, but showed a random opposite trend or even irregular fluctuation in the short term [20]. Therefore, false alarms or missing alarms were prone to occur, which challenged accuracy and stability of fire perception systems. Besides, the methods based on Bayesian estimation, statistics, and inference had low computational complexity, but the coupling relationship among the sensors was not adequately considered. Simultaneously, the fire perception systems using the methods based on artificial intelligence had excellent stability and robustness, but detection speed of the systems ought to be improved due to the complexity of algorithm calculation. Finally, the difficulty of fire experiments and high risk factors led to the lack of relevant sensor data, which made the development of deep learning algorithms requiring a large amount of fire sensor data slow [22].

The characteristic data captured by the sensors in a fire are essentially time-series data [23]. Bai et al. [24] proposed TCN by improving the special structure on the basis of a convolutional neural network (CNN). TCN has been shown to outperform common deep learning models in processing time-series data in multiple domains, such as natural language processing [25], traffic flow prediction [26], sound event localization and detection (SELD) [27], and fault diagnosis [28]. After extracting sensor data features, the TCN connected to the Softmax function through the fully connected (FC) layer for prediction. What is more, the Softmax function simply converted the features processed by the FC layer into probability distributions. Thus, classification accuracy was affected by the features without great linear separability.

In conclusion, aiming at the shortcomings of existing fire classification algorithms, a multi-sensor fusion indoor fire perception algorithm named TCN-AAP-SVM was proposed in this work. First of all, sensor data were preprocessed. Preprocessing contained data cleaning, data filtering, trend extraction, and sliding window processing. Compared with the BP neural network mentioned in the literature, the time dimension information was sufficiently considered by the methods of trend extraction and sliding window in this work. Then, the features of processed data were fully extracted by improved TCN. Based on the TCN in the literature, a new residual block was designed to meet the requirements of fast feature extraction and accurate classification of fire perception. What is more, the AAP layer was utilized to reduce the dimension of the extracted features and improve detection speed of the subsequent classifier. Compared with the FC layer, no parameter needed to be optimized in the AAP layer. Thus, the speed of the training and detection was improved. Finally, the fire classification was realized by the SVM classifier with a Gaussian radial basis kernel function instead of the Softmax function. The SVM classifier had excellent nonlinear classification ability because of the soft interval maximization classification method.

## 2. Materials and Methods

The structure of the TCN-AAP-SVM proposed in this work is presented in Figure 1. TCN-AAP-SVM is composed of the input layer, improved TCN layer, the AAP layer, and the SVM classifier. The input layer is used for data preprocessing, starting with cleaning and filtering of the raw data. Then, the trend of the filtered sensor data is extracted by the Mann–Kendall algorithm. Moreover, the data is cut with length n by the sliding window method. The output of the input layer consists of filtered and trend data of $C_{i}\;$ sensors, with the size of $2\times C_{i}\times n$. The current and n−1 historical data of sensors are handled by the improved TCN layer which consists of several residual blocks. The channel count of the hidden layer in the improved TCN layer is $C_{h}$, and the output size of the improved TCN layer is $C_{h}\times n$. The training of TCN-AAP-SVM is based on the backpropagation and optimization of the Adam optimizer until loss convergence stops. The feature size extracted from the improved TCN layer can be transformed into 1×3 according to the category count in the AAP layer, which simplifies the calculation of the subsequent classifier when analyzing feature data. Finally, the fire classification results are worked out according to the output of the AAP layer by the SVM classifier. The proposed algorithm in this work extracts the feature data sufficiently and reduces structural redundancy in TCN. Therefore, it is expected to achieve excellent performance in fire perception classification.

### 2.1. Trend Extraction

Trend extraction is the key point of data preprocessing in the input layer. The trend data of the sensor, including direction and magnitude, are placed into the improved TCN layer together with the sensor data. The Mann–Kendall algorithm is used for trend extraction in this work, which is simple to calculate and suitable for fire scenes requiring fast perception. The trend statistic (*S*) of time-series data $P=\left\{x_{1},x_{2},\ldots ,x_{T}\right\}$ is described as Equations (1) and (2):(1)$$S=\overset{T-1}{\underset{l=1}{\sum }}\overset{T}{\underset{k=l+1}{\sum }}sgn\left(x_{k}-x_{l}\right)$$
(2)$$\mathrm{sgn}\left(x_{k}-x_{l}\right)=\{\begin{array}{c} 1\;\;\;\;,x_{k}-x_{l}>0\\ 0\;\;\;\;,x_{k}-x_{l}=0\\ -1\;\;\;,x_{k}-x_{l}<0 \end{array}$$
where T is the length of time-series data, and S contains T(T−1)/2 trend components. The trend components are normalized to [−1, 1] to obtain the trend value τ representing the entire time-series data. The trend is increasing when τ>0, and decreasing when τ<0. The closer τ approaches 1 or −1, the greater the trend. The calculation of τ is shown in Equation (3):(3)$$\tau =\frac{2S}{T\left(T-1\right)}$$

### 2.2. Improved TCN Layer

The improved TCN layer contains several residual blocks. As shown in Figure 2, a new residual block structure is designed. The new residual block includes a dilated causal convolution and non-linear layer. A leaky Relu activation function is used in the non-linear layer, which is mathematically expressed as Equation (4):(4)$$y_{i}=\{\begin{array}{c} x_{i},\;|\;x_{i}\geq 0\\ \frac{x_{i}}{a_{i}},\;|\;x_{i}<0 \end{array},i=1,2,\ldots ,m$$
where $a_{i}$ is a fixed parameter in the range of (1,+∞).

The leaky Relu assigns a non-zero slope to the value of the negative half axis based on the Relu activation function. Reasonable selection of $a_{i}$ increases the fitting ability of the improved TCN layer near zero, thus the risk of overfitting is reduced effectively. Besides, weight normalization and dropout are added to the new residual block for regularization to avoid gradient disappearance and gradient explosion during the training. The features at both endpoints of the residual connection are added directly unless the sizes of the features are different. To account for discrepancy in the sizes of the features, an additional convolution with s=1×1 is used to ensure the residual connection features are of the same size.

The position diagram of the new residual block in the improved TCN layer is exhibited in Figure 3. The receptive field of the improved TCN layer is related to the hidden layer depth, kernel size, and expansion coefficient. The relationship between expansion coefficient d and hidden layer depth j is described as $d=2^{j}$. Therefore, the receptive field increases exponentially with the increase of the hidden layer depth. The size of the receptive field is expressed as (s−1)×d. Since the convolution is causal, the output at moment n of the j layer is only determined by the data from moment n to moment n−d×s of the j−1 layer. Thus, the future information in the data cannot be disclosed, which causes the proposed algorithm to have the potential to be applied to real-time perception systems. Moreover, the computation is greatly reduced due to the interval sampling and weight sharing of dilated convolution.

### 2.3. The AAP Layer

The AAP layer is an average pooling layer with adaptive dimension, which is used to replace the FC layer for dimension reduction and improving detection speed of the subsequent classifier. The kernel size $S_{k}$ and stride $S_{t}$ of pooling are adaptively calculated according to input size $S_{i}$ and output size $S_{o}$ of the AAP layer, as shown in Equations (5) and (6):(5)$$S_{k}=S_{i}-\left(S_{o}-1\right)\times S_{t}$$
(6)$$S_{t}=\mathrm{floor}\left(S_{i}-S_{o}\right)$$
where floor() is round down calculation.

The output size of the AAP layer is relative to the number of categories. In the m classification problem, m pooling cores are used to perform the average pooling process for the characteristic output of the improved TCN layer. Due to no parameters needing to be optimized in the AAP layer, the speed of the training and detection is improved. What is more, the risk of over-fitting in training is reduced.

### 2.4. The SVM Classifier

It is difficult to distinguish sensor data between smoldering that has just occurred and no-fire. Therefore, the part with linear inseparability in the output features of the AAP layer limits classification accuracy. The SVM classifier, instead of the Softmax function, is used for improving classification accuracy of the fire perception system. The SVM classifier has better non-linear classification performance than the Softmax function in handling multi-classification tasks with small samples [29]. The SVM classifier constructs the maximum soft interval separation hyperplane in the high-dimensional feature space. Moreover, Gaussian radial basis function (GRBF) and soft interval maximization are used for binary classification of the feature data extracted from the above hidden layers. Then, the binary classification is extended to multiple classification by treating each class as a positive set and the other classes as negative sets, respectively. Finally, the classification probability of each positive category is compared, and the maximum corresponds to the final classification result. For the sample set $Z=\left\{\left(x_{1},y_{1}\right),\left(x_{2},y_{2}\right),\ldots ,\left(x_{n},y_{n}\right)\right\}$, the objective function of the SVM classifier is demonstrated as Equations (7) and (8):(7)$$\underset{w,b,\xi }{\min}\frac{1}{2}\|w^{2}\|+c\overset{n}{\underset{i=1}{\sum }}\xi_{i},\;\xi_{i}>0$$
(8)$$s.t.y_{i}\left(w^{T}\varphi \left(x_{i}\right)+b\right)\geq 1-\xi_{i},i=1,2,\ldots ,n$$
where $\xi_{i}$ is the relaxation variable which is shown as Equation (9), and $\varphi \left(x_{i}\right)$ is the formulation of GRBF shown as Equation (10):(9)$$\xi_{i}=\max\left(0,1-y_{i}\left(w^{T}\varphi \left(x_{i}\right)+b\right)\right)>0$$
(10)$$\varphi \left(x_{i}\right)=e^{\left(-\frac{\left\|x_{i}-l_{i}\right\|}{2\sigma^{2}}\right)}$$
where w and b are optimization parameters, which are adjusted to minimize the value of the objective function to maximize the soft interval segmentation surface.

### 2.5. Data Set and Data Preprocessing

The effectiveness of the proposed algorithm in this work is tested on the open-source home smoke alarm test data set of the National Institute of Standards and Technology (NIST) [30]. The NIST conducted a series of experiments in residential buildings to investigate the performance of different sensors in fire detection and alarms. The ignition chamber included a bedroom, living room, and kitchen. Sensors were arranged in the ignition chamber, such as temperature, smoke, CO, CO_2_, oxygen, ions smoke alarm, photoelectric smoke alarm, and so on. The plan of the experimental site is shown in Figure 4. All experiment equipment was corrected twice before the start of all experiments, and recalibration was performed in the fire emulator/detector evaluator (FE/DE) before the start of each experiment. The FE/DE is a single-channel wind tunnel that meet all the conditions required to evaluate the performance of point-type sensors. The experimental data had high reliability due to the above operations.

In this work, the training set contains 16 experimental data, and the test set contains 8 experimental data. The experimental situation of the training set and the test set is summarized as shown in Table 1 and Table 2. It should be pointed out that the bedroom door was closed in SDC09, SDC14, and SDC36, while the bedroom door was open in the rest of the experiments.

As shown in Table 1 and Table 2:(1)The training data overview: The training data is count is 2483, containing 979 no-fire data, 448 flaming data, and 1056 smoldering data.(2)The test data overview: The test data count is 800, containing 392 no-fire data, 152 flaming data, and 256 smoldering data.

In view of the above data, the following processing is carried out in each experiment:(1)Data cleaning: The temperature, smoke, and CO data are selected from more than 100 sensors in each experiment. In addition, the current data and historical data are very inconsistent with the no-fire characteristics of each experiment before ignitions are deleted.(2)Data filtering: The Kalman filter algorithm is used to suppress and attenuate the noise in the sensor data.(3)Trend extraction: The Mann–Kendall algorithm is used to extract the trend of real-time data and 15 historical data in the three sensors. Note: The initial 15 data of each experiment are stable no-fire data, thus the corresponding trend value is determined to be 0.(4)Sliding window processing: After the above processing, the data contains temperature, smoke, and CO in each experiment, as well as the corresponding trend values. Then the data required for TCN-APP-SVM is constructed by the sliding window method. The window size and step size are set to 20 and 10. Therefore, each data contains current data and historical data in the time dimension. The processed results are displayed in Figure 5. The data are tagged as no-fire, flaming, and smoldering according to the type of fire experiment and when the first sensor responded to the fire [31]. After the normalization of the data in Figure 5, data preprocessing is completed.

## 3. Results

### 3.1. Evaluation Index

The performance of algorithms was evaluated by indexes in this work, which included accuracy, precision, recall, *F*1 score, receiver operating characteristic (ROC) curve, and area under ROC curve (AUC). *Accuracy* is the percentage of the correctly predicted samples in the total samples in the whole test set, which intuitively reflects the overall classification ability of algorithms. Precision is the percentage of the correctly predicted positive samples to the predicted positive samples, which can accurately reflect accuracy of each category of prediction results. Recall is the percentage of positive samples correctly predicted in actual positive samples, which can represent the probability of each category predicted. *F*1 score is the harmonic average of precision and recall, which can characterize the comprehensive performance measurement of algorithms in precision and recall well. The value of the preceding indicators ranges from 0 to 1. The closer the indicator is to 1, the better the performance is. The indexes are shown as Equations (11)–(14):(11)$$Accuracy=\frac{TN+TP}{TN+FP+TP+FN}\;$$
(12)$$Precision=\frac{TP}{TP+FP}\;$$
(13)$$Recall=\frac{TP}{\left(TP+FN\right)}\;$$
(14)$$F1\;score=\frac{2\times Precision\times Recall}{Precision+Recall}\;$$
where TP is the number of correctly classified positive samples, TN is the number of correctly classified negative samples, FP is the number of incorrectly classified positive samples, and FN is the number of incorrectly classified negative samples.

The vertical axis of ROC curve is true positive rate (*TPR*). *TPR* represents the proportion of the predicted positive and actually positive samples to all positive samples, as shown in Equation (15). The horizontal axis of ROC curve is false positive rate (*FPR*). FPR stands for the proportion of positive predicted and actually negative samples in all negative samples, as shown in Equation (16). The ROC curve can embody the classification performance of algorithms at each sample point. The closer the curve is to the top left, the better the performance, which is not affected by the imbalance of samples. *AUC*, which can more accurately reflect the overall numerical classification ability of the algorithms, is generally calculated by the statistical method displayed in Equation (17).
(15)$$TPR=\frac{TP}{TP+FN}$$
(16)$$FPR=\frac{FP}{FP+TN}$$
(17)$$AUC=\frac{\sum_{i\in positive}rank_{i}-\frac{M\left(1+M\right)}{2}}{M\times N}$$
where *M* is the count of positive samples and *N* is the count of negative samples.

### 3.2. Contrast Experiment

The algorithms in the work were implemented on a Dell Inspiron-3558 laptop equipped with an i5-4210U@1.70GHz CPU and 8 GB of operating memory. The algorithms were trained and tested in PyCharm. The experimental results were the average of 20 repeated experiments.

In the NIST open-source home smoke alarm test data set, TCN-AAP-SVM was compared with TCN, BP neural network, and LSTM. The settings of the adjustable parameters of TCN-AAP-SVM are shown in Table 3.

The settings of the adjustable parameters of TCN are the same as TCN-AAP-SVM. The network structure and the settings of the adjustable parameters of the BP neural network and LSTM are the same as [2,21]. Accuracy, training time, and test time of each comparative algorithm are shown in Table 4.

Table 4 shows that accuracy of TCN-APP-SVM is improved by 2.5%, 8.95%, and 2.75%, respectively, compared with TCN, BP neural network, and LSTM. The corresponding training speed is improved by 49.72%, 37.58%, and 50.51%, respectively. The corresponding detection speed is improved by 46.39%, 52.55%, and 15.52%, respectively. It can be seen that TCN-APP-SVM is more effective than the comparison algorithms in classification ability and operation speed. The multi-category confusion matrix is utilized to further quantify the count and position of correct and incorrect classification of each algorithm, as shown in Figure 6.

The horizontal axis represents the classification results of fire perception, and the vertical axis represents the actual label in Figure 6. The test data for no-fire, flaming, and smoldering are 390, 152, and 256, respectively. The no-fire test data is slightly different from the statistical data in Table 2 due to the existence of batch size. The numbers on the main diagonal represent the data correctly classified for each category of fire perception. However, it is worth mentioning that the test data for no-fire, flaming, and smoldering of the BP neural network are 4042, 1516, and 2557, respectively, because the BP neural network can only process single point data without the sliding window method. Precision, recall, and *F*1 score of each class can be calculated from the confusion matrix of each algorithm, as shown in Table 5. ROC curve and *AUC* of each algorithm are shown in Figure 7.

As can be seen from Table 5, TCN-AAP-SVM has the highest precision for no-fire, reaching up 97.49%. TCN has a slightly higher precision for no-fire than LSTM, while the BP neural network has the lowest precision, only 81.37%. However, there is no significant difference in no-fire recall for each algorithm. According to the comprehensive analysis of no-fire *F*1 score, the no-fire classification ability of TCN-AAP-SVM is superior to all comparison algorithms.

Flaming precision of each algorithm is basically the same, but there is a great difference in flaming recall. Although flaming recall of TCN-AAP-SVM is 7.56% lower than the BP neural network, it is 4% higher than TCN and LSTM. According to the comprehensive analysis of flaming *F*1 score, the flaming classification ability of the proposed algorithm is better than TCN and LSTM, even if it is inferior to the BP neural network.

The BP neural network also has good performance in smoldering precision, but smoldering recall is unacceptably poor, at only 64.45%. However, TCN-AAP-SVM shows the most outstanding performance, reaching up to 98.05%. According to the comprehensive analysis of smoldering *F*1 score, TCN-AAP-SVM has the most excellent smoldering classification ability, while the smoldering classification ability of the BP neural network is far inferior to TCN and LSTM.

ROC curve of TCN-AAP-SVM is closest to the upper left corner as Figure 7 shows, which indicates that the overall classification effect is remarkable. Meanwhile, the overall classification effects of TCN and LSTM are highly similar, which are more excellent than the BP neural network. The value of AUC accurately quantifies the above analysis. AUC of TCN-AAP-SVM is 98.42%, slightly higher than that of TCN and LSTM, while the BP neural network has the lowest AUC of 96.71%. Therefore, the proposed algorithm in this work has the most superior classification performance overall compared with other comparison algorithms.

### 3.3. Ablation Study

Ablation experiments are carried out to explore the influence of different classifier selections, the AAP layer, and different adjustable parameter selections on the classification performance of the proposed algorithm. In order to explore the influence of different classifier selections on the proposed algorithm, five classifiers are selected for comparison with the SVM classifier, containing the Softmax function, multi-layer perceptron (MLP), Gaussian naive Bayes (GNB), K-nearest neighbor (KNN), and random forest (RF). The comparison results are shown in Table 6.

As Table 6 shows, the SVM classifier has the highest classification accuracy, reaching up to 97.49%, but the corresponding detection speed is inferior to other classifiers except KNN. Meanwhile, detection speed of KNN is the slowest, but the corresponding classification accuracy is medium. Classification accuracy of RF is second only to KNN, and the corresponding detection speed is slightly higher than the SVM classifier. Softmax function, MLP, and GNB have similar detection speeds, while GNB has the lowest classification accuracy among all kinds of classifiers.

In order to explore the influence of the AAP layer on the proposed algorithm, the performance of the proposed algorithm with AAP layer and FC layer is explored. The results are shown in Table 7.

As is shown in Table 7, compared with TCN-AAP-SVM, the classification accuracy and the detection speed of TCN-FC-SVM decrease by 0.75% and 12.09%, respectively. Therefore, the classified results of the model without the AAP layer become worse.

In order to explore the influence of different adjustable parameter selections on the proposed algorithm, a series of experiments are conducted to analyze four selected parameters. To begin with, the influence of different window size and step size selections in the sliding window method on classification accuracy of the proposed algorithm are explored. The window size and step size intervals are 5. Window size ranges from 10 to 20, and step size ranges from 5 to 25. The results are shown in Figure 8.

It can be seen from Figure 8 that high classification accuracy appears when step size ranges from 10 to 15 and window size ranges from 20 to 25. The area is surrounded by low elevations, which shows that classification accuracy will decline when window size or step size is too large or too small. It is worth mentioning that no matter what the window size is, classification accuracy will be greatly reduced if the step size is less than 10. To sum up, it is vital to choose appropriate window size and step size.

Moreover, different hidden layer depths and channel counts are set in the training and test process to explore the influence of different choices of hidden layer depth and channel count on classification accuracy of the proposed algorithm. The interval of hidden layer depth is 1, and the value ranges from 2 to 6. Channel count increases by a multiple of 2, and the value ranges from 6 to 96. However, when channel count is 96, the network model is enormous, which violates the principle of fast and accurate detection. Therefore, the maximum channel count is 64 instead of 96.

It can be seen from Figure 9 that classification accuracy is the highest when hidden layer depth is 4 and channel count is 24, reaching up to 97.49%. Classification accuracy from that point in all directions decreases, but at different speeds in different directions. The two directions with the fastest speed are the directions in which hidden layer depth and channel count increase or decrease simultaneously. Therefore, network models that are too complex or too simple cannot achieve excellent classification accuracy. However, no matter which parameter combination is selected, classification accuracy of the proposed algorithm is higher than that of the comparison algorithms.

## 4. Discussion

As Table 4 shows, accuracy, training speed, and detection speed of TCN-AAP-SVM are superior compared with the comparison algorithms in this work. The promotion of speed is owing to improved TCN structure and the addition of the AAP layer. The promotion of classification accuracy is on account of the trend extraction and sliding window method, which fully consider the time dimension information of sensor data. In addition, the excellent non-linear classification ability of the SVM classifier also makes a great contribution to the improvement of classification accuracy. As displayed in Table 5, the probability of smoldering samples predicted by the BP neural network is far less than other comparison algorithms, because the BP neural network cannot process current data and historical data simultaneously through the sliding window method. Besides, smoldering burns quite slowly, resulting in insignificant changes in sensor data during the initial stage of the fire. Therefore, the BP neural network without considering the time dimension information of sensor data has difficulty distinguishing the no-fire and smoldering samples.

The SVM classifier selected in this work leads to the highest classification accuracy as displayed in Table 6, which confirms that the SVM classifier has significant classification performance in small fire sample perception. Although detection speed of the SVM classifier is not the fastest, it is not important compared with the significant improvement of the calculation speed of the proposed algorithm by improved TCN structure. Besides, as is shown in Table 7, the classified results of the model without the AAP layer become worse. A large amount of data will increase the difficulty for the SVM classifier to find the optimal soft interval classification surface. The function of the AAP layer is to reduce 3D features to 2D and reduce the size of features as much as possible on the premise of retaining the acquired features, in order that SVM classifier can achieve better classification effect.

In a certain range, no matter what the window size is, classification accuracy will be greatly reduced if the step size is less than 10 as exhibited in Figure 8. This is mainly because even if the time dimension information is taken into account, too small a step size will still reduce the differences among samples of different categories, resulting in classification difficulties.

As demonstrated in Figure 9, when the hidden layer depth and channel count of the network model decrease simultaneously, classification accuracy decreases rapidly. This is mainly caused by the problem of insufficient feature extraction ability of an overly simple network model. Besides, when the hidden layer depth and channel count increase simultaneously, accuracy decreases rapidly as well. This shows that blindly increasing the complexity of the network model cannot effectively improve the classification performance.

## 5. Conclusions and Future Work

Indoor fires cause huge casualties and economic losses worldwide. Aiming at the shortcomings of existing fire classification algorithms, TCN-AAP-SVM was proposed in this work. The analysis in this work leads to the following conclusions. Firstly, the time dimension information of sensor data is taken fully into account by the methods of trend extraction and sliding window in this work. Therefore, the proposed algorithm has the best excellent classification performance among comparison algorithms, especially the ability to distinguish between no-fire samples and smoldering samples. Secondly, the increase in detection speed is owed mainly to improved TCN structure. Huge improvement in feature extraction speed can make the classifier selection only pay attention to the classification ability and ignore the impact of speed to optimize the fire perception system. Finally, the appropriate selection of adjustable parameters is vital to the improvement of classification accuracy. The classification accuracy is excellent under most parameter selections, which proves the stability of the proposed algorithm.

In multi-sensor fire perception, the selection of sensor combination is seldom paid attention to. However, the selection of sensor combination with low redundancy and high relevance can further improve the classification accuracy of fire perception system. Therefore, further research on the selection of sensor combination in fire perception issue is warranted.

## Figures and Tables

**Figure 1 sensors-22-04550-f001:**
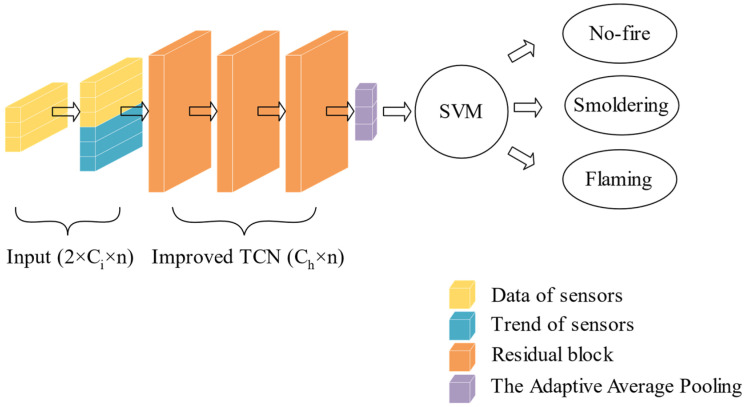
The structure of TCN-AAP-SVM.

**Figure 2 sensors-22-04550-f002:**
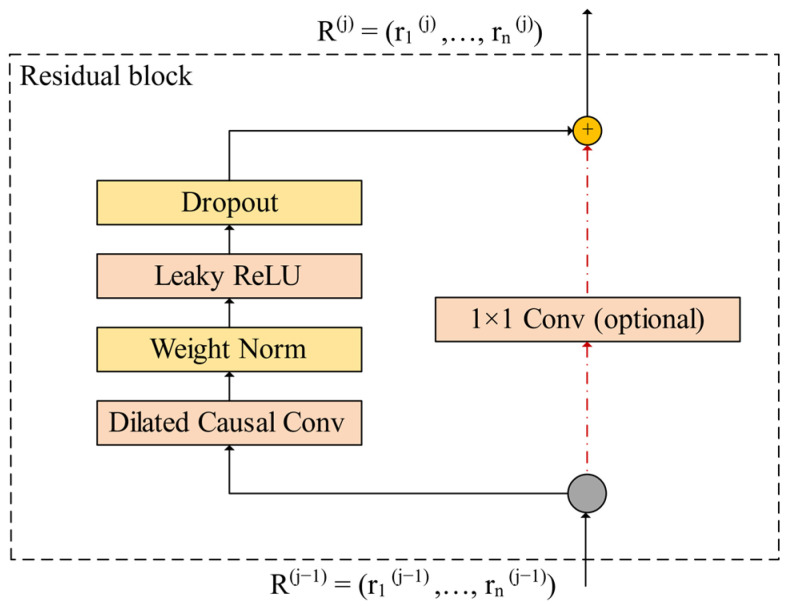
The structure of the new residual block.

**Figure 3 sensors-22-04550-f003:**
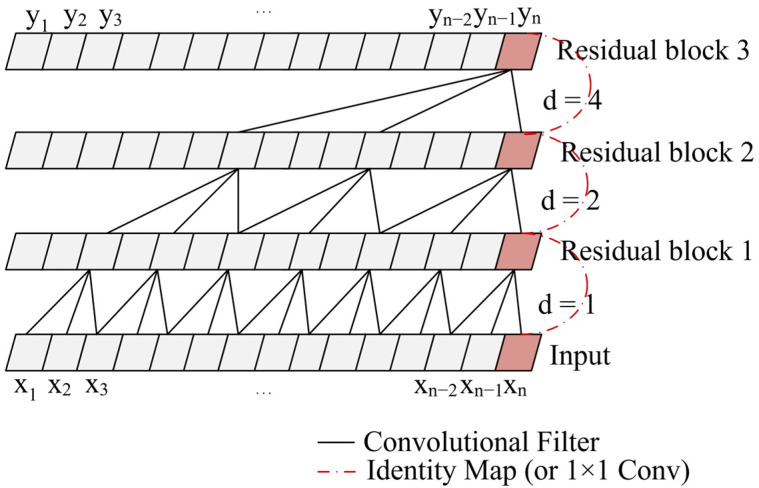
The structure of improved TCN with kernel size s=1×3 .

**Figure 4 sensors-22-04550-f004:**
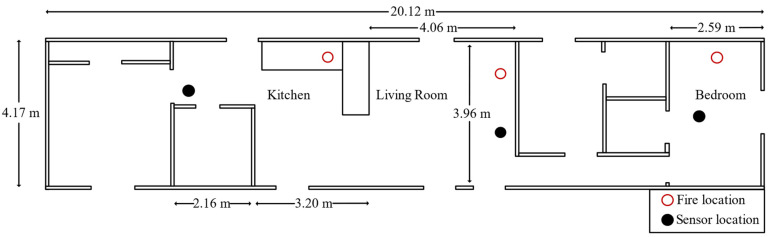
The experimental site plan of NIST.

**Figure 5 sensors-22-04550-f005:**
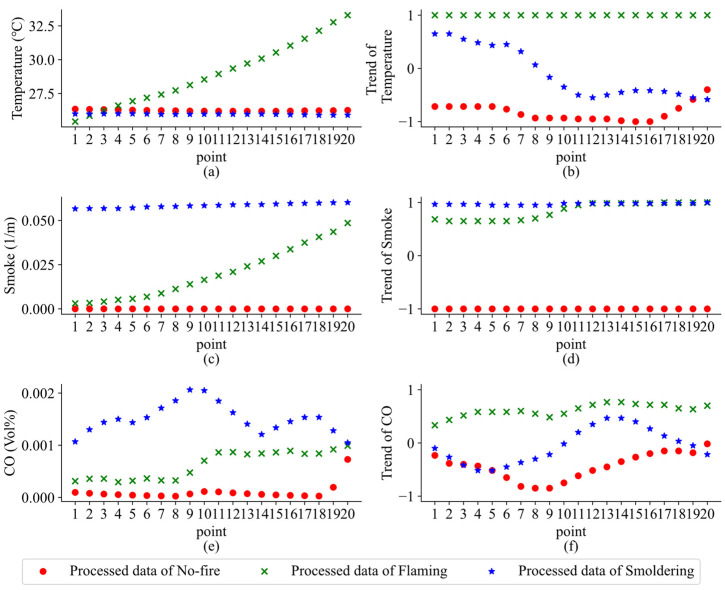
The processed data of (**a**) temperature, (**b**) temperature trend, (**c**) smoke, (**d**) smoke trend, (**e**) CO, and (**f**) CO trend.

**Figure 6 sensors-22-04550-f006:**
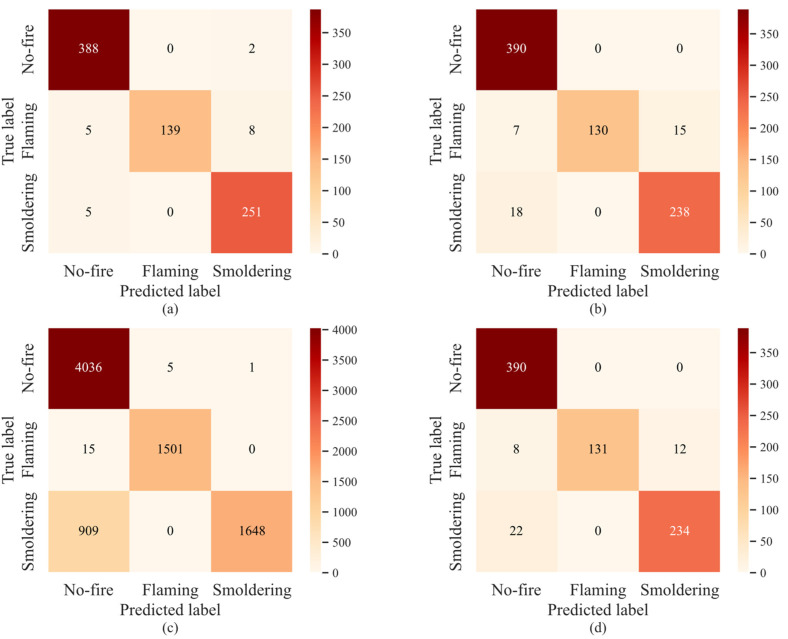
The confusion matrix of (**a**) TCN-AAP-SVM, (**b**) TCN, (**c**) BP neural network, and (**d**) LSTM.

**Figure 7 sensors-22-04550-f007:**
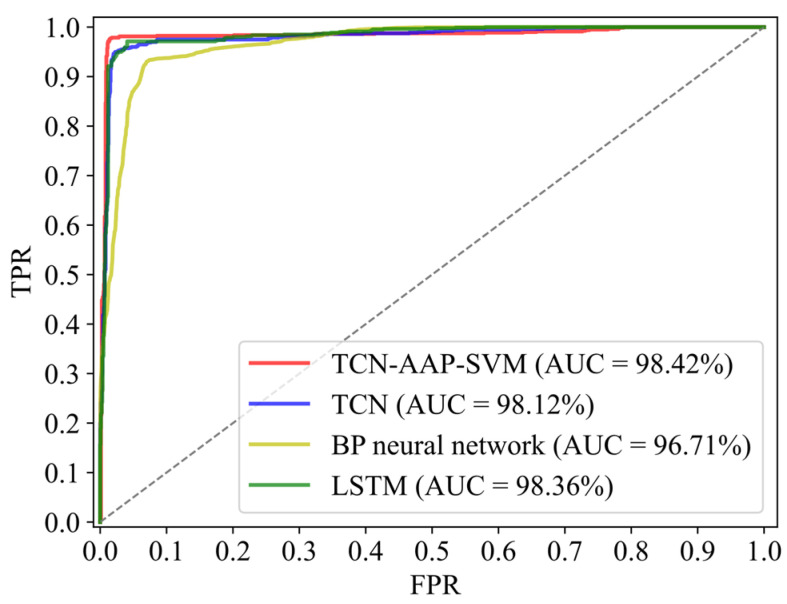
ROC curve and AUC of each comparison algorithm.

**Figure 8 sensors-22-04550-f008:**
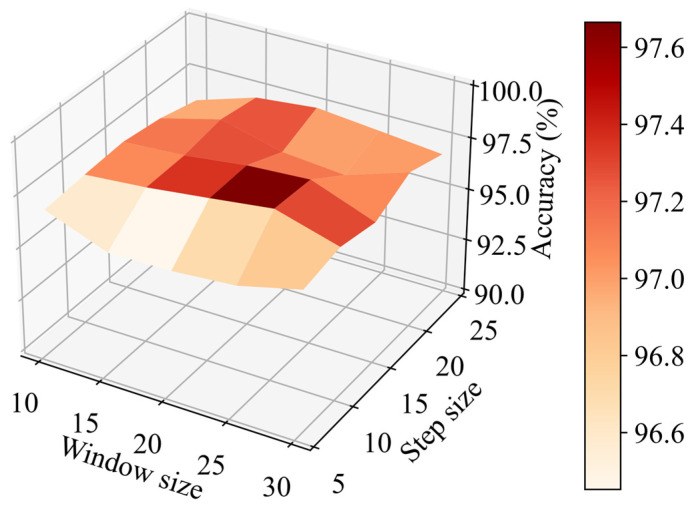
Classification accuracy of the proposed algorithm with different selections of window size and step size.

**Figure 9 sensors-22-04550-f009:**
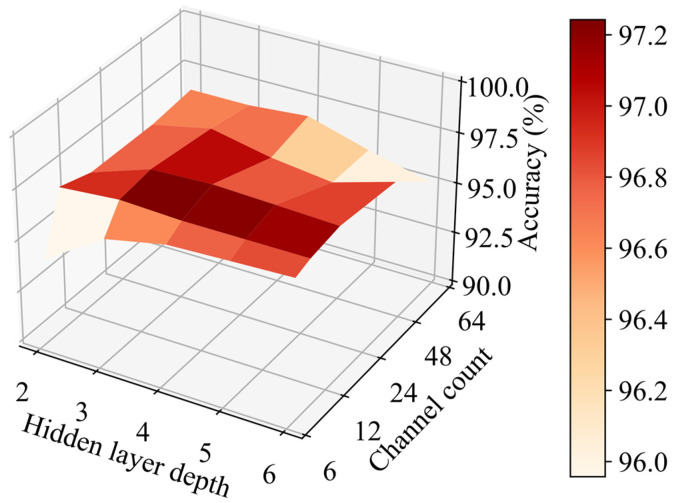
Classification accuracy of the proposed algorithm with the different choices of hidden layer depth and channel count.

**Table 1 sensors-22-04550-t001:** The experimental situation of the training set.

Scenario	Fire Type	Fire Material	Fire Location	Division Time (s)	No-Fire Data Count	Fire Data Count
SDC02	Flaming	Chair	Living area	84	13	15
SDC04	Smoldering	Mattress	Bedroom	8	13	98
SDC05	Flaming	Mattress	Bedroom	87	10	6
SDC08	Smoldering	Mattress	Bedroom	131	19	177
SDC11	Smoldering	Chair	Living area	53	30	209
SDC12	Flaming	Oil	Kitchen	233	48	71
SDC13	Flaming	Oil	Kitchen	100	33	88
SDC31	Smoldering	Chair	Living area	4931	270	158
SDC34	Smoldering	Chair	Living area	218	40	184
SDC35	Flaming	Chair	Living area	94	63	8
SDC36	Flaming	Mattress	Bedroom	32	145	98
SDC37	Smoldering	Mattress	Bedroom	130	70	88
SDC38	Flaming	Mattress	Bedroom	35	60	59
SDC39	Flaming	Mattress	Bedroom	22	85	5
SDC40	Smoldering	Mattress	Bedroom	522	45	142
SDC41	Flaming	Oil	Kitchen	110	35	98

**Table 2 sensors-22-04550-t002:** The experimental situation of the test set.

Scenario	Fire Type	Fire Material	Fire Location	Division Time (s)	No-Fire Data Count	Fire Data Count
SDC01	Smoldering	Chair	Living area	377	16	138
SDC06	Smoldering	Mattress	Bedroom	83	9	118
SDC07	Flaming	Mattress	Bedroom	59	33	18
SDC09	Flaming	Mattress	Bedroom	31	36	56
SDC10	Flaming	Chair	Living area	100	50	17
SDC14	Flaming	Mattress	Bedroom	3398	193	41
SDC15	Flaming	Chair	Living area	271	27	13
SDC33	Flaming	Chair	Living area	88	28	7

**Table 3 sensors-22-04550-t003:** The setting of adjustable parameters of TCN-AAP-SVM.

Adjustable Parameters	Value
Batch size	3
Train epoch	30
Dropout	0.5
Kernel size	1×3
Expansion coefficient	2
Window size	20
Step size	10
Hidden layer depth	4
Channel count	24
Optimizer	Adam
Learning rate	0.002
Punish coefficient	0.1
Kernel function	RBF

**Table 4 sensors-22-04550-t004:** The performance of each comparison algorithm.

Performance	TCN-AAP-SVM	TCN	BP Neural Network	LSTM
Accuracy	97.49	94.99	88.54	94.74
Train time (s)	289.6270	576.0081	1025.4182	584.0498
Test time (s)	1.1643	2.1718	2.4536	1.3782

**Table 5 sensors-22-04550-t005:** The evaluation indexes of each comparison algorithm.

Evaluation Index	TCN-AAP-SVM	TCN	BP Neural Network	LSTM
No-fire precision	97.49	93.98	81.37	92.86
Flaming precision	100	100	99.67	100
Smoldering precision	96.17	94.07	99.94	95.12
No-fire recall	99.49	100	99.85	100
Flaming recall	91.45	85.83	99.01	86.75
Smoldering recall	98.05	92.97	64.45	91.41
No-fire *F*1 score	98.48	96.90	89.67	96.30
Flaming *F*1 score	95.53	92.20	99.34	92.90
Smoldering *F*1 score	97.10	93.52	78.36	93.23

**Table 6 sensors-22-04550-t006:** The performance of the proposed algorithm with different classifiers.

Performance	SVM	Softmax	MLP	GNB	KNN	RF
Accuracy	97.49	96.99	96.87	96.12	96.62	96.37
Test time (s)	1.1533	1.0194	1.0214	1.0204	1.2842	1.1263

**Table 7 sensors-22-04550-t007:** The performance of the proposed algorithm with AAP layer and full connected layer.

Performance	TCN-AAP-SVM	TCN-FC-SVM
Accuracy	97.49	96.74
Test time (s)	1.1533	1.2927

## Data Availability

Publicly available datasets were analyzed in this study. These data can be found here: https://www.nist.gov/el/nist-report-test-fr-4016 (accessed on 23 April 2022). The code and data generated from this study are openly available at https://github.com/yuukijiang/Fire_perception (accessed on 23 April 2022).

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
