# Peer review of "Research on Multi-Sensor Fusion Indoor Fire Perception Algorithm Based on Improved TCN"

_sensors, 2022, doi:10.3390/s22124550_

Round 1
Reviewer 1 Report
This paper provides an indoor fire perception algorithm based on multi-sensor fusion. The algorithm combines TCN and Pooling with SVM training the data set which has three sensors' outputs and their trends. The simulated results show that the algorithm has better classified results than others. Upon the paper, the following items have to clarify in the proper place in the paper.
1. Are the new built feathers, call the trend features, very important to the algorithm?
2. Can the new built feathers replace the original sensor output? If not, why the algorithm uses the AAP to reduce the data dimensions?
3. If the AAP is removed from the model, the classified results may become worse?
4.More important is that the model will be changing with time. To get good classified reresults along the time, How the model re-train?
5. What is the 'Channel count' in Table 3, which is set to 24?
Author Response
The authors would like to thank you for the time and effort spent in reviewing the manuscript.
According to your suggestions and opinions, we have made explanations and improvements one by one.
Please see the attachment.

Reviewer 2 Report
This paper discusses a an indoor fire perception algorithm based on multi-sensor fusion. In the first stage, the authors use an improved Temporal Convolutional Network to extract the features of the data collected by the sensors. Then, Adaptive Average Pooling (AAP) is used to reduce the dimensions of the extracted features. Finally, a Support Vector Machine (SVM) classifier is developed to perform the classification.
- The paper's structure is sound and clear. The topic discussed is relevant to the journal's topics of interest. Additionally, the degree of novelty in this work is good
- The evaluation is thorough and the results (which are based on a real-world dataset) are very encouraging towards the efficiency of the proposed solution.
Some minor comments from my side, which I believe that would contribute to making the presentation even better, would be:
- The part of the Introduction which is dedicated to the paper's contributions needs strengthening. The authors should explain in more detail how their work is different than the work in the literature, as well as give more information on every part of their novelty. Maybe some parts of the Discussion Section could be reused.
- In the related work part of the Introduction and specifically in part (1), (Bayesian estimation, statistics, and inference), the authors could include also some other recent works on the field which make use of Edge Computing technology to deal with rapid fire detection:
[1] Avgeris, M., Spatharakis, D., Dechouniotis, D., Kalatzis, N., Roussaki, I., & Papavassiliou, S. (2019). Where there is fire there is smoke: A scalable edge computing framework for early fire detection. Sensors, 19(3), 639.
[2] Maltezos, E., Petousakis, K., Dadoukis, A., Karagiannidis, L., Ouzounoglou, E., Krommyda, M., ... & Amditis, A. (2022). A Smart Building Fire and Gas Leakage Alert System with Edge Computing and NG112 Emergency Call Capabilities. Information, 13(4), 164.
- In the Conclusions section, the authors are strongly advised to not use a list-like structure for their findings. Also, some discussion on their future work is always more than welcome!
Author Response
The authors would like to thank you for the time and effort spent in reviewing the manuscript. According to your opinion, we modify the introduction and conclusion section in the paper.
Please see the attachment.

Round 2
Reviewer 1 Report
All attention I paid have been explained clearly, and the authors have improved their manuscription properly.